# Improving the informativeness of Mendelian disease-derived pathogenicity scores for common disease

Samuel S. Kim [1,2✉], Kushal K. Dey [2], Omer Weissbrod [2], Carla Márquez-Luna[3], Steven Gazal [2] & Alkes L. Price [2,4,5✉]

Despite considerable progress on pathogenicity scores prioritizing variants for Mendelian disease, little is known about the utility of these scores for common disease. Here, we assess the informativeness of Mendelian disease-derived pathogenicity scores for common disease and improve upon existing scores. We first apply stratified linkage disequilibrium (LD) score regression to evaluate published pathogenicity scores across 41 common diseases and complex traits (average $N = 320K$). Several of the resulting annotations are informative for common disease, even after conditioning on a broad set of functional annotations. We then improve upon published pathogenicity scores by developing AnnotBoost, a machine learning framework to impute and denoise pathogenicity scores using a broad set of functional annotations. AnnotBoost substantially increases the informativeness for common disease of both previously uninformative and previously informative pathogenicity scores, implying that Mendelian and common disease variants share similar properties. The boosted scores also produce improvements in heritability model fit and in classifying disease-associated, fine-mapped SNPs. Our boosted scores may improve fine-mapping and candidate gene discovery for common disease.

[1] Department of Electrical Engineering and Computer Science, Massachusetts Institute of Technology, Cambridge, MA 02142, USA. [2] Department of Epidemiology, Harvard T.H. Chan School of Public Health, Boston, MA 02115, USA. [3] Charles Bronfman Institute for Personalized Medicine, Icahn School of Medicine at Mount Sinai, New York, NY 10029, USA. [4] Program in Medical and Population Genetics, Broad Institute of MIT and Harvard, Cambridge, MA 02142, USA. [5] Department of Biostatistics, Harvard T.H. Chan School of Public Health, Boston, MA 02115, USA. ✉email: sungil@mit.edu; aprice@hsph.harvard.edu

Despite considerable progress on pathogenicity scores prioritizing both coding and non-coding variants for Mendelian disease[1–10] (reviewed in ref. [11]), little is known about the utility of these pathogenicity scores for common disease. The shared genetic architecture between Mendelian disease and common disease has been implicated in studies reporting the impact of genes underlying monogenic forms of common diseases on the corresponding common diseases[12], significant comorbidities among Mendelian and complex diseases[13], and gene-level overlap between Mendelian diseases and cardiovascular diseases[14–16], neurodevelopmental traits[17,18], and other complex traits[19]. However, variant-level assessment of shared genetic architecture using Mendelian disease-derived pathogenicity scores has not been explored. Thus, our current understanding of the genetic relationship between Mendelian disease and common disease remains limited.

Here, we assess the informativeness of Mendelian disease-derived pathogenicity scores for common disease and improve upon existing scores. We focus our attention on polygenic common and low-frequency variant architectures, which explain the bulk of common disease heritability[20–24]. We assess the informativeness of annotations defined by top variants from published Mendelian disease-derived pathogenicity scores by applying stratified linkage disequilibrium (LD) score regression[25] (S-LDSC) with the baseline-LD model[26,27] to 41 independent common diseases and complex traits (average $N = 320$ K). We assess informativeness conditional on the baseline-LD model, which includes a broad set of coding, conserved, regulatory, and LD-related annotations. Then, we improve upon the published pathogenicity scores by developing AnnotBoost, a gradient boosting-based machine-learning framework to impute and denoise pathogenicity scores using functional annotations from the baseline-LD model. We assess the informativeness of annotations defined by top variants from the boosted scores by applying S-LDSC to quantify conditional informativeness after considering annotations from the baseline-LD model as well as annotations derived from the corresponding published scores. We also assess the informativeness of the published and boosted pathogenicity scores in producing improvements in heritability model fit and in predicting disease-associated, fine-mapped SNPs.

We find that several annotations derived from published pathogenicity scores are informative for common disease, even after conditioning on annotations from the baseline-LD model. Furthermore, AnnotBoost substantially increases the informativeness for a common disease of both previously uninformative and previously informative pathogenicity scores, implying that Mendelian and common disease variants share similar properties. We conclude that our boosted scores have high potential to improve fine-mapping and candidate gene discovery for common disease.

## Results

### Overview of methods

We define a binary annotation as an assignment of a binary value to each of low-frequency ($0.5\% \leq$ MAF $< 5\%$) and common (MAF $\geq 5\%$) SNP in a 1000 Genomes Project European reference panel[28], as in our previous work[25,27]. We define a pathogenicity score as an assignment of a numeric value quantifying predicted pathogenicity, deleteriousness, and/or protein function to some or all of these SNPs; we refer to theses score as Mendelian disease-derived pathogenicity scores, as these scores have predominantly been developed and assessed in the context of Mendelian disease (e.g., using pathogenic variants from ClinVar[29] and HGMD[30]). We analyze 11 Mendelian disease-derived missense scores, six genome-wide Mendelian disease-derived scores, and 18 additional scores. Our primary focus is on

binary annotations defined either using top variants from published (missense or genome-wide) Mendelian disease-derived pathogenicity scores, or using top variants from boosted scores that we constructed from those pathogenicity scores using AnnotBoost, a gradient boosting-based framework that we developed to impute and denoise pathogenicity scores using 75 codings, conserved, regulatory and LD-related annotations from the baseline-LD model[26,27] (Supplementary Fig. 1; see "Methods"). AnnotBoost uses decision trees to distinguish pathogenic variants (defined using the input pathogenicity score) from benign variants; the AnnotBoost model is trained using the XGBoost gradient boosting software[31]. AnnotBoost uses odd (respectively even) chromosomes as training data to make predictions for even (respectively odd) chromosomes; the output of AnnotBoost is the predicted probability of being pathogenic. We note that Mendelian disease-derived pathogenicity scores may score a subset of SNPs, but every baseline-LD model annotation scores all SNPs. Further details are provided in the Methods section; we have publicly released open-source software implementing AnnotBoost (see "Code availability"), as well as all pathogenicity scores and binary annotations analyzed in this work (see "Data availability").

We assessed the informativeness of the resulting binary annotations for common disease heritability by applying S-LDSC[25] to 41 independent common diseases and complex traits[32] (average $N = 320$ K; Supplementary Table 1; see "Data availability"), conditioned on coding, conserved, regulatory and LD-related annotations from the baseline-LD model[26,27] and meta-analyzing results across traits. We assessed informativeness for common disease using standardized effect size ($\tau^*$), defined as the proportionate change in per-SNP heritability associated to a one standard deviation increase in the value of the annotation, conditional on other annotations[26] (see "Methods"). We also computed the heritability enrichment, defined as the proportion of heritability divided by the proportion of SNPs. Unlike enrichment, $\tau^*$ quantifies effects that are unique to the focal annotation; annotations with significantly positive or negative $\tau^*$ are informative after considering all other annotations in the model, whereas annotations with $\tau^* = 0$ contain no unique information, even if they are enriched for heritability (see "Methods"). While S-LDSC models linear combinations of functional annotations, AnnotBoost constructs (linear and) non-linear combinations of baseline-LD model annotations to provide unique information.

### Informativeness of Mendelian disease-derived missense scores for common disease

We assessed the informativeness for a common disease of binary annotations derived from 11 Mendelian disease-derived pathogenicity scores for missense variants[1,5–8,33–37] (see Table 1). These scores reflect the predicted impact of missense mutations on Mendelian disease; we note that our analyses of the common disease are focused on common and low-frequency variants, but these scores were primarily trained using very rare variants from ClinVar[29] and Human Gene Mutation Database (HGMD)[30]. For each of the 11 missense scores, we constructed binary annotations based on top missense variants using five different thresholds (from top 50% to top 10% of missense variants) and applied S-LDSC[25,26] to 41 independent common diseases and complex traits (Supplementary Table 1), conditioning on coding, conserved, regulatory and LD-related annotations from the baseline-LD model[26,27] and meta-analyzing results across traits; proportions of top SNPs were optimized to maximize informativeness (see "Methods"). We incorporated the 5 different thresholds into the number of hypotheses tested when assessing statistical significance (Bonferroni $P < 0.05/500 = 0.0001$, based on a total

**Table 1 11 Mendelian disease-derived missense and six genome-wide Mendelian disease-derived pathogenicity scores.**

| Score | Description | Coverage (% SNPs scored) | Ref. |
|---|---|---|---|
| PolyPhen-2 | Impact of missense variants using protein sequence and structure using HumDiv | 0.28% | 1,33 |
| PolyPhen-2-HVAR | Impact of missense variants using protein sequence and structure using HumVar | 0.28% | 1,33 |
| MetaLR | Deleterious missense mutations using ensemble scoring (logistic regression) | 0.32% | 34 |
| MetaSVM | Deleterious missense mutations using ensemble scoring (support vector machine) | 0.32% | 34 |
| PROVEAN | Impact of an amino acid change on protein function | 0.31% | 35,82 |
| SIFT 4G | Impact of an amino acid change on protein function | 0.31% | 5 |
| REVEL | Pathogenic missense variants using ensemble scoring | 0.32% | 6 |
| M-CAP | Pathogenic rare missense variants | 0.03% | 7 |
| PrimateAI | Impact of missense variants using deep neural networks | 0.26% | 8 |
| MPC | Regional missense constraint | 0.10% | 36 |
| MVP | Impact of missense variants using deep neural networks | 0.29% | 37 |
| CADD | Predicted deleterious variants using ensemble scoring | 100% | 2,46 |
| Eigen | Putatively causal variants using unsupervised learning | 83.79% | 3 |
| Eigen-PC | Putatively causal variants using unsupervised learning using the lead eigenvector | 83.79% | 3 |
| ReMM | Pathogenic regulatory variants using ensemble scoring | 100% | 4 |
| NCBoost | Pathogenic non-coding variants using ensemble scoring | 28.55% | 10 |
| ncER | Essential regulatory variants using ensemble scoring | 61.94% | 9 |

For each of 17 Mendelian disease-derived pathogenicity scores analyzed, we provide a description and report the coverage (% of SNPs scored) and corresponding reference. The first 11 annotations are scores for missense variants, and the last 6 annotations are genome-wide scores. Annotations are ordered first by type and then by the year of publication.

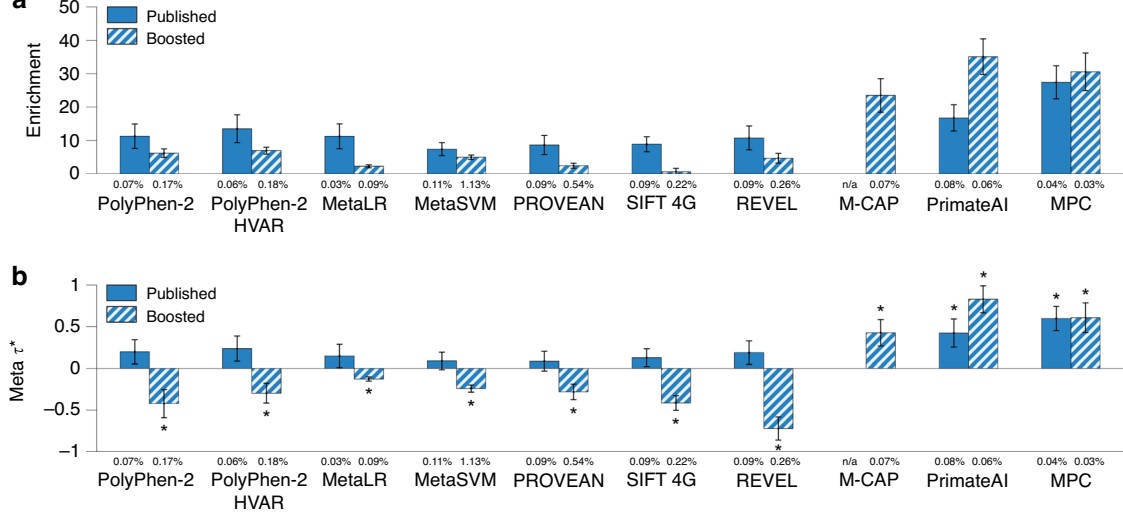

**Fig. 1 Informativeness for a common disease of binary annotations derived from 11 Mendelian disease-derived missense scores and corresponding boosted scores.** We report (**a**) heritability enrichment of binary annotations derived from published and boosted Mendelian disease-derived missense scores, meta-analyzed across 41 independent traits; **b** conditional $\tau^*$ values, conditioning on the baseline-LD model (for annotations derived from published scores) or the baseline-LD model and corresponding published annotations (for annotations derived from boosted scores). We report results for 10 Mendelian disease-derived missense scores (of 11 analyzed) for which annotations derived from published and/or boosted scores were conditionally significant; the published M-CAP score spanned too few SNPs to be included in the S-LDSC analysis. The percentage under each bar denotes the proportion of SNPs in the annotation; the proportion of top SNPs included in each annotation was optimized to maximize informativeness (largest $|\tau^*|$ among Bonferroni-significant annotations, or most significant p-value if no annotation was Bonferroni-significant). Error bars denote 95% confidence intervals. In (**b**), * denotes conditionally significant annotations. Numerical results are reported in Supplementary Data 1. Results for standardized enrichment, defined as enrichment times the standard deviation of annotation value (to adjust for annotation size), are reported in Supplementary Data 23.

of ≈ 500 hypotheses tested in this study; see "Methods"). We identified (Bonferroni-significant) conditionally informative binary annotations derived from two published missense scores: the top 30% of SNPs from MPC[36] (enrichment = 27x (s.e. 2.5), $\tau^*$ = 0.60 (s.e. 0.07)) and the top 50% of SNPs from PrimateAI[8] (enrichment = 17x (s.e. 2.0), $\tau^*$ = 0.42 (s.e. 0.09)) (Fig. 1, Table 2 and Supplementary Data 1). The MPC (Missense badness, PolyPhen-2, and Constraint) score[36] is computed by identifying

regions within genes that are depleted for missense variants in ExAC data[38] and incorporating variant-level metrics to predict the impact of missense variants; the PrimateAI score[8] is computed by eliminating common missense variants identified in other primate species (which are presumed to be benign in humans), incorporating a deep-learning model trained on the amino acid sequence flanking the variant of interest and the orthologous sequence alignments in other species. The remaining

**Table 2 Summary of informativeness for a common disease of annotations derived from 82 published scores and corresponding boosted scores.**

| Score | # Scores | # Marginally significant annotations | | # Significant annotations in a combined joint model | |
|---|---|---|---|---|---|
| | | Published | Boosted | Published | Boosted |
| Mendelian missense | 11 | 2[a] | 10 | 1[a] | 2 |
| Genome-wide Mendelian | 6 | 3 | 6 | 2 | 3 |
| Additional scores | 18 | 6[b] | 13 | 0[b] | 0 |
| Baseline-LD model annotations | 47 | n/a | 24 | n/a | 3 |

For each category of scores, we report the number of scores; the number of marginally conditionally informative annotations (S-LDSC $\tau^* p < 0.0001$, conditional on the baseline-LD model) (baseline-LD + marginal model); and the number of jointly conditionally informative annotations in a combined joint model (S-LDSC $\tau^* p < 0.0001$ and $|\tau^*| \geq 0.25$, conditional on the baseline-LD model and each other) (baseline-LD + joint model).
[a]Based on 9/11 published Mendelian missense scores analyzed, as binarized annotations were too small to analyze for the remaining two published Mendelian missense scores.
[b]Based on 16/18 published additional scores analyzed, as binarized annotations from the remaining two published additional scores had proportions of SNPs < 0.02%.

published Mendelian disease-derived missense scores all had derived binary annotations that were significantly enriched for disease heritability (after Bonferroni correction) but not conditionally informative (except for the published M-CAP[7] score, which spanned too few SNPs to be included in the S-LDSC analysis).

We constructed boosted scores from the 11 Mendelian disease-derived missense scores using AnnotBoost, a gradient boosting-based machine-learning framework that we developed to impute and denoise pathogenicity scores using functional annotations from the baseline-LD model[26] (see "Methods"). We note that AnnotBoost scores genome-wide (missense and non-missense) variants, implying low genome-wide correlations between input Mendelian disease-derived missense scores and corresponding genome-wide boosted scores (0.0–0.24; Supplementary Data 2A). AnnotBoost attained high predictive accuracy in out-of-sample predictions of input missense scores (AUROC = 0.76–0.94, AUPRC = 0.43–0.82; Supplementary Data 3), although we caution that high predictive accuracy does not necessarily translate into conditional informativeness for common disease[39]. We further note that out-of-sample AUROCs closely tracked the genome-wide correlations between input Mendelian disease-derived missense scores and corresponding genome-wide boosted scores ($r = 0.65$), implying that accurately predicting the input pathogenicity scores results in more correlated boosted scores.

For each missense pathogenicity score, after running Annot-Boost, we constructed binary annotations based on top genome-wide variants, using 6 different thresholds (ranging from top 10% to top 0.1% of genome-wide variants, as well as variants with boosted scores ≥0.5; see "Methods"). We assessed the informativeness for the common disease of binary annotations derived from each of the 11 boosted scores using S-LDSC, conditioning on annotations from the baseline-LD model and 5 binary annotations derived from the corresponding published Mendelian disease-derived missense score (using all 5 thresholds) (baseline-LD + 5). We identified conditionally informative binary annotations derived from boosted versions of 10 Mendelian disease-derived missense scores, including eight previously uninformative scores and the two previously informative scores (Fig. 1, Table 2 and Supplementary Data 1). Letting ↑ denote boosted scores, examples include the top 0.1% of SNPs from M-CAP↑[7], a previously uninformative score (enrichment = 23x (s.e. 2.6), $\tau^* = 0.43$ (s.e. 0.08); the published M-CAP pathogenicity score spanned too few SNPs to be included in the S-LDSC analysis of Fig. 1) and the top 0.1% of SNPs from PrimateAI↑[8], a previously informative score (enrichment = 35x (s.e. 2.7), $\tau^* = 0.83$ (s.e. 0.08)). The M-CAP (Mendelian Clinically Applicable Pathogenicity) score[7] is computed by training a gradient boosting tree classifier to distinguish pathogenic variants from HGMD[30]

vs. benign variants from ExAC[38] using 9 pathogenicity likelihood scores as features (including PolyPhen-2[1], MetaLR[34], CADD[2]; see Table 1); the PrimateAI score is described above. Interestingly, binary annotations derived from 7 boosted scores had significantly negative $\tau^*$ ($-0.72$ (s.e. 0.07) to $-0.13$ (s.e. 0.01)). All but one of these binary annotations were significantly enriched for disease heritability, but less enriched than expected based on annotations from the baseline-LD + 5 model (Supplementary Table 2; see ref. [40] and "Methods", resulting in significantly negative $\tau^*$. These annotations are thus conditionally informative for disease heritability (analogous to transposable element annotations in ref. [40] that were significantly depleted, but less depleted than expected and thus conditionally informative); as noted above, annotations with significantly positive or negative $\tau^*$ are conditionally informative. The boosted version of the remaining Mendelian disease-derived missense score (MVP↑; not included in Fig. 1) had a derived binary annotation that was significantly enriched for disease heritability (after Bonferroni correction) but not conditionally informative (Supplementary Data 1).

We performed five secondary analyses. First, we restricted the 10 significant binary annotations derived from our boosted Mendelian disease-derived missense scores to non-coding regions, which were previously unscored by the Mendelian disease-derived missense scores, and assessed the informativeness of the resulting non-coding binary annotations using S-LDSC. We determined that the non-coding annotations retained the bulk of the overall signals (85–110% of absolute $\tau^*$; Supplementary Data 4), implying that AnnotBoost leverages information about pathogenic missense variants to usefully impute scores for non-missense variants. Second, we investigated which features of the baseline-LD model contributed the most to the informativeness of the boosted annotations by applying shapley additive explanation (SHAP)[41], a widely used tool for interpreting machine-learning models. We determined that conservation-related features drove the predictions of the boosted annotations, particularly (binary and continuous) GERP scores[42] (Supplementary Fig. 2). Third, we examined the trait-specific S-LDSC results, instead of meta-analyzing results across 41 traits. We identified two (respectively 32) annotation-trait pairs with significant $\tau^*$ values for annotations derived from published (respectively boosted) scores (Supplementary Data 5); significance was assessed using FDR <5%, as Bonferroni correction would be overly conservative in this case. These annotation-trait pairs spanned 1 (respectively 10) different annotations, all of which were also conditionally significant in the meta-analysis across traits (Fig. 1b). Fourth, we assessed the heterogeneity of heritability enrichment and conditional informativeness ($\tau^*$) across the 41 traits (as in ref. [23]; see "Methods"). We determined

that 10/20 annotations tested had significant heterogeneity in enrichment and 14/20 annotations tested had significant heterogeneity in $\tau^*$, implying substantial heterogeneity across traits (Supplementary Data 6). Fifth, we investigated the overlap between genes linked to each of our 12 conditionally significant annotations and 165 gene sets of biological importance, including high-pLI genes[38] and known Mendelian genes[19] (see Methods; Supplementary Data 7 and 8). We linked SNPs to the nearest gene, scored genes based on the maximum pathogenicity score of linked SNPs, and assessed overlap between top gene score quintiles and each of 165 reference gene sets. We consistently observed excess overlap for genes for which homozygous knockout in mice results in lethality[43,44] and high-pLI genes[38], and depleted overlap for olfactory receptor genes[45] (Supplementary Data 9A, B). In addition, top gene score quintiles derived from our boosted scores often had significantly more overlap (based on Fisher's exact test) than corresponding top gene quintiles derived from published scores (Supplementary Data 10; results for PolyPhen-2[1,33] shown in Supplementary Fig. 3); gene scores derived from published and boosted pathogenicity scores were moderately correlated (Supplementary Data 9C, D). This implies that our new annotations can help identify biologically important genes.

We conclude that two Mendelian disease-derived missense annotations and 10 boosted annotations are conditionally informative for common disease, relative to baseline-LD model annotations.

**Informativeness of genome-wide Mendelian disease-derived pathogenicity scores for common disease.** We assessed the informativeness for a common disease of binary annotations derived from six genome-wide Mendelian disease-derived pathogenicity scores[2–4,9,10] (see Table 1). These scores reflect the predicted impact of (coding and) non-coding variants on Mendelian disease; as above, these scores were primarily trained using very rare variants from ClinVar[29] and HGMD[30]. For each of the 6 genome-wide scores, we constructed binary annotations based on top genome-wide variants using five different thresholds (from top 0.1% to top 10% of genome-wide variants) and applied S-LDSC to the 41 traits, conditioning on the baseline-LD model[26] and meta-analyzing results across traits; proportions of top SNPs were optimized to maximize informativeness (see "Methods"). We identified (Bonferroni-significant) conditionally informative binary annotations derived from three genome-wide scores: the top 0.5% of SNPs from ReMM[4] (enrichment = 19x (s.e. 1.2), $\tau^*$ = 0.82 (s.e. 0.09)), the top 0.5% of SNPs from CADD[2,46] (enrichment = 18x (s.e. 1.3), $\tau^*$ = 0.71 (s.e. 0.10)), and the top 0.1% of SNPs from Eigen[3] (enrichment = 24x (s.e. 2.1), $\tau^*$ = 0.40 (s.e. 0.06)) (Fig. 2, Table 2 and Supplementary Data 11). The CADD (Combined Annotation Dependent Depletion) score[2,46] is computed by training a support vector machine to distinguish deleterious vs. neutral variants using functional annotations as features; the Eigen score[3] is computed from 29 input functional annotations by using an unsupervised machine-learning method (leveraging blockwise conditional independence between annotations) to differentiate functional vs. non-functional variants; the ReMM (Regulatory Mendelian Mutation) score[4] is computed by training a random forest classifier to distinguish 406 hand-curated Mendelian mutations from neutral variants using conservation scores and functional annotations as features. The remaining three genome-wide scores all had derived binary annotations that were significantly enriched for disease heritability (after Bonferroni correction) but not conditionally informative (Supplementary Data 11).

We applied AnnotBoost to the 6 genome-wide Mendelian disease-derived scores. We observed moderate correlations between input genome-wide Mendelian disease-derived scores and corresponding boosted scores ($r = 0.35–0.66$; Supplementary Data 2B). AnnotBoost again attained high predictive accuracy in out-of-sample predictions of input genome-wide scores (AUROC = 0.83–1.00, AUPRC = 0.70–1.00; Supplementary Data 3); however, out-of-sample AUROCs did not closely track the correlations between input genome-wide scores and corresponding boosted scores ($r = 0.05$).

We again constructed binary annotations based on top genome-wide variants, using six different thresholds (ranging from top 0.1% to top 10% of genome-wide variants, as well as variants with boosted scores $\geq 0.5$; see "Methods"). We assessed the informativeness for a common disease of binary annotations derived from each of the six boosted scores using S-LDSC, conditioning on annotations from the baseline-LD model and five binary annotations derived from the corresponding published genome-wide Mendelian disease-derived score (using all five thresholds). We identified conditionally informative binary annotations derived from boosted versions of all six genome-wide Mendelian disease-derived scores, including the three previously uninformative scores and the three previously informative scores (Fig. 2, Table 2 and Supplementary Data 1). Examples include the top 5% of SNPs from ncER↑[9] (enrichment = 6.2x (s.e. 0.30), $\tau^*$ = 0.74 (s.e. 0.10)) and the top 0.5% of SNPs from boosted Eigen-PC↑[3] (enrichment = 16x (s.e. 1.1), $\tau^*$ = 0.62 (s.e. 0.12)), both of which were previously uninformative scores, and the top 1% of SNPs from ReMM↑[4] (enrichment = 17x (s.e. 0.8), $\tau^*$ = 1.17 (s.e. 0.12)), a previously informative score. The ncER (non-coding Essential Regulation) score[9] is computed by training a gradient boosting tree classifier to distinguish non-coding pathogenic variants from ClinVar[29] and HGMD[30] vs. benign variants using 38 functional and structural features; the Eigen-PC score[3] (related to the Eigen score) is computed from 29 input functional annotations by using the lead eigenvector of the annotation covariance matrix to weight the annotations; the ReMM score is described above.

We performed five secondary analyses. First, for the four genome-wide Mendelian disease-derived scores with <100% of SNPs scored (Table 1), we restricted the binary annotations derived from our boosted genome-wide Mendelian disease-derived scores to previously unscored variants and assessed the informativeness of the resulting binary annotations using S-LDSC. We determined that these annotations retained only a minority of the overall signals (17–54% of absolute $\tau^*$; Supplementary Data 12), implying that AnnotBoost usefully denoises previously scored variants. Second, we again investigated which features of the baseline-LD model contributed the most to the informativeness of the boosted annotations by applying SHAP[41]. We determined that both conservation-related features (e.g. GERP scores) and LD-related features (e.g. LLD-AFR; the level of LD in Africans) drove the predictions of the boosted annotations (Supplementary Fig. 4). Third, we examined the trait-specific S-LDSC results, instead of meta-analyzing results across 41 traits. We identified 11 (respectively 20) annotation-trait pairs with significant $\tau^*$ values for annotations derived from published (respectively boosted) scores (FDR < 5%; Supplementary Data 5). These annotation-trait pairs spanned 2 (respectively 4) different annotations, all of which were also conditionally significant in the meta-analysis across traits (Fig. 2b). Fourth, we assessed the heterogeneity of heritability enrichment and $\tau^*$ across the 41 traits (see Methods). We determined that 10/12 annotations tested had significant heterogeneity in enrichment and 9/12 annotations tested had significant heterogeneity in $\tau^*$, implying substantial heterogeneity across traits (Supplementary Data 6). Fifth, we investigated the overlap between genes linked to each of our nine conditionally significant annotations and 165 gene sets of biological importance (see "Methods"; Supplementary Data 7). As above, we consistently

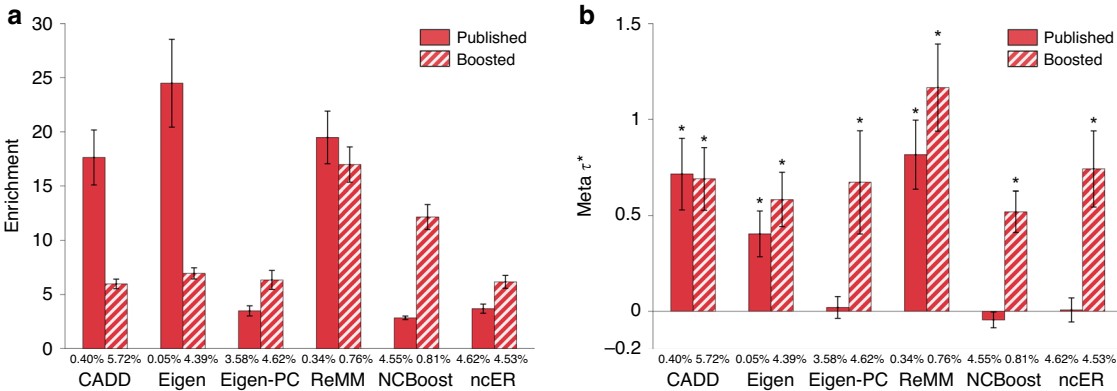

**Fig. 2 Informativeness for a common disease of binary annotations derived from 6 genome-wide Mendelian disease-derived scores and corresponding boosted scores.** We report (**a**) heritability enrichment of binary annotations derived from published and boosted genome-wide Mendelian disease-derived scores, meta-analyzed across 41 independent traits; **b** conditional $\tau^*$ values, conditioning on the baseline-LD model (for annotations derived from published scores) or the baseline-LD model and corresponding published annotations (for annotations derived from boosted scores). We report results for six genome-wide Mendelian disease-derived scores (of six analyzed) for which annotations derived from published and/or boosted scores were conditionally significant. The percentage under each bar denotes the proportion of SNPs in the annotation; the proportion of top SNPs included in each annotation was optimized to maximize informativeness (largest $|\tau^*|$ among Bonferroni-significant annotations, or top 5% if no annotation was Bonferroni-significant; top 5% was the average optimized proportion among significant annotations). Error bars denote 95% confidence intervals. In panel (**b**), * denotes marginally conditionally significant annotations. Numerical results are reported in Supplementary Data 11. Results for standardized enrichment, defined as enrichment times the standard deviation of annotation value (to adjust for annotation size), are reported in Supplementary Data 23.

observed excess overlap for genes for which homozygous knockout in mice results in lethality[43,44] and high-pLI genes[38], and depleted overlap for olfactory receptor genes[45] (Supplementary Data 9), implying that our new annotations can help identify biologically important genes. For example, the top (bottom) quintile of genes linked to the boosted CADD annotation had 1.7x (0.34x) excess overlap with high-pLI genes[38] vs. excess overlap of 0.8x (0.5x) for the top (bottom) quintile of genes linked to the published CADD[2,46] annotation (OR = 2.4, $P < 6e-49$) (Supplementary Fig. 3, Supplementary Data 10).

We conclude that three genome-wide Mendelian disease-derived annotations and six boosted annotations are conditionally informative for common disease, relative to baseline-LD model annotations.

**Informativeness of additional genome-wide scores for common disease.** For completeness, we assessed the informativeness for a common disease of 18 additional genome-wide scores not related to Mendelian disease, including two constraint-related scores[47,48], nine scores based on deep-learning predictions of epigenetic marks[49–51], and seven gene-based scores[38,52–54] (see Supplementary Data 13). For each of the 18 additional scores, we constructed binary annotations based on top variants using five different thresholds and applied S-LDSC to the 41 traits, conditioning on the baseline-LD model[26] and meta-analyzing results across traits; in this analysis, we also conditioned on eight Roadmap annotations[55] (four annotations based on the union across cell types and four annotations based on the average across cell types, as in ref. [39]), as many of the additional scores pertain to regulatory elements, making this an appropriate conservative step.

We identified (Bonferroni-significant) conditionally binary annotations derived from six informative scores, including the top 1% of SNPs from CDTS[47] (enrichment = 9.3x (s.e. 0.75), $\tau^* = 0.35$ (s.e. 0.06)) and the top 5% of SNPs from DeepSEA-H3K4me3[49,50] (enrichment = 3.9x (s.e. 0.23), $\tau^* = 0.21$ (s.e. 0.04)) (Fig. 3, Table 2 and Supplementary Data 14). CDTS (Context-Dependent Tolerance Score)[47] is a constraint score based on observed vs. expected variation in whole-genome sequence data; DeepSEA-H3K4me3 scores[49,50] are computed by training a deep-learning model to

predict chromatin marks using proximal reference genome sequence as features and aggregated across different cell types[39] (The DeepSEA annotations in Fig. 3 were more significant than those analyzed in ref. [39], because we optimized binary annotations based on top variants; however, no DeepSEA annotations were included in our combined joint model (see below)). Nine of the remaining 10 scores (excluding two that were not analyzed due to small annotation size) had derived binary annotations that were significantly enriched for disease heritability (after Bonferroni correction) but not conditionally informative (Supplementary Data 14).

We applied AnnotBoost to the 18 additional scores, and to the 47 main annotations of the baseline-LD model (Supplementary Data 13). Correlations between input scores and corresponding boosted scores varied widely ($r = 0.005–0.93$; Supplementary Data 2C). AnnotBoost again attained high predictive accuracy in out-of-sample predictions of the input scores (AUROC = 0.55–1.00, AUPRC = 0.23–0.98; Supplementary Data 3); out-of-sample AUROCs closely tracked the correlations between input scores and corresponding boosted scores ($r = 0.65$).

We again constructed binary annotations based on top genome-wide variants, using six different thresholds (ranging from top 0.1% to top 10% of genome-wide variants, as well as variants with boosted scores $\geq 0.5$; see "Methods"). We assessed the informativeness for a common disease of binary annotations derived from each of the 65 boosted scores using S-LDSC, conditioning on annotations from the baseline-LD model, the 8 Roadmap annotations, and (for the first 18 additional scores only) five binary annotations derived from the corresponding input scores (using all five thresholds). We identified conditionally informative binary annotations derived from boosted versions of 13/18 additional scores (including 11 previously uninformative scores and two previously informative scores) and 24/47 baseline-LD model annotations (Fig. 3, Table 2 and Supplementary Data 14). Examples include the top 10% of SNPs from DeepSEA-DNase↑[49,50] (enrichment = 3.7x (s.e. 0.27), $\tau^* = 0.69$ (s.e. 0.11)), a previously uninformative score, the top 1% of SNPs from CCR↑[48] (enrichment = 7.9x (s.e. 0.65), $\tau^* = 0.51$ (s.e. 0.09)), a previously uninformative score, and the top 5% of SNPs from H3K9ac↑[56] (enrichment = 5.4x (s.e. 0.31), $\tau^* = 0.76$ (s.e. 0.09)), a

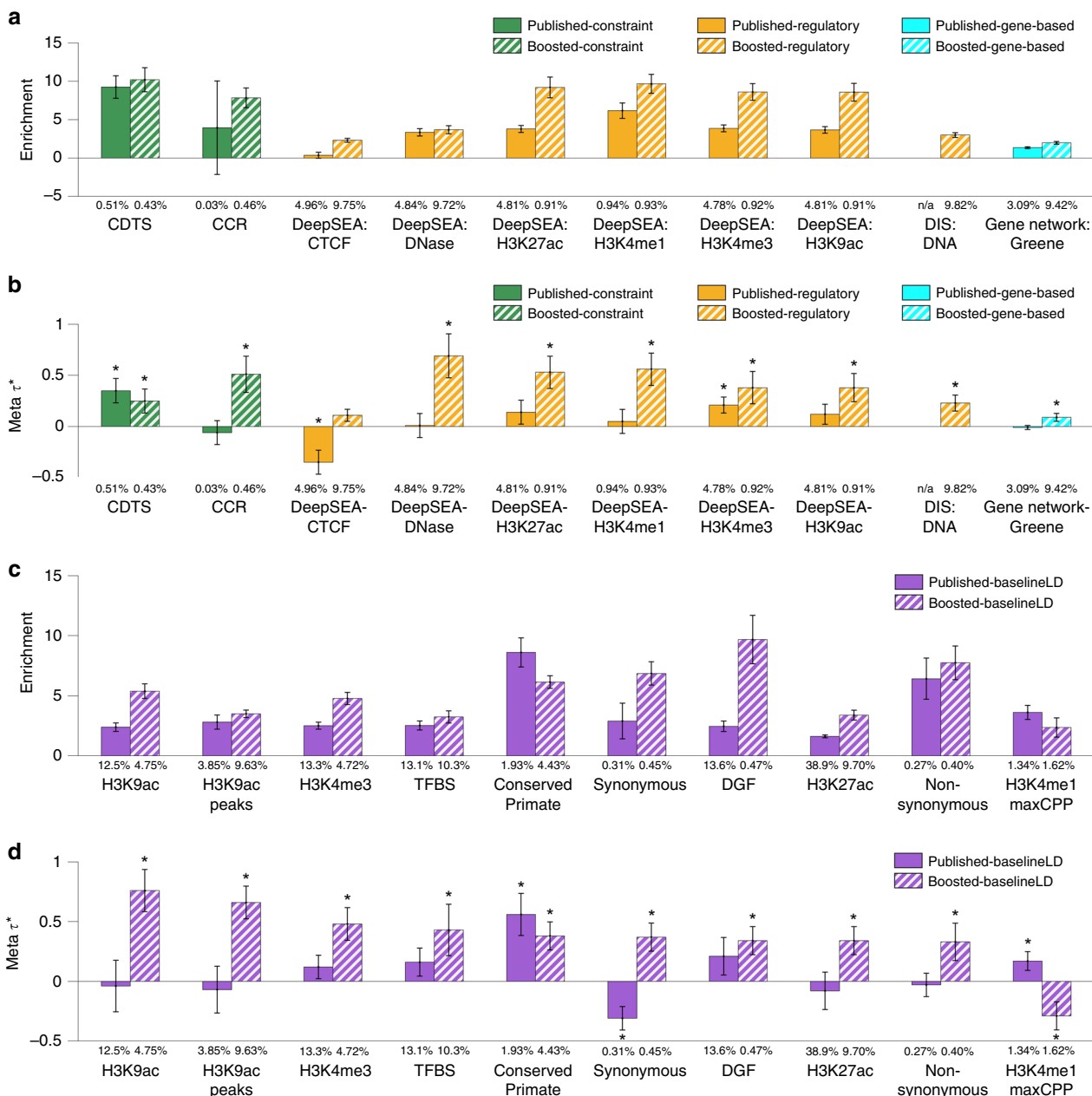

**Fig. 3 Informativeness for a common disease of binary annotations derived from 18 additional genome-wide scores + 47 baseline-LD model annotations and corresponding boosted scores.** We report (**a**) heritability enrichments of binary annotations derived from published and boosted additional genome-wide scores, meta-analyzed across 41 independent traits; (**b**) conditional $\tau^*$ values, conditioning on the baseline-LD model and eight Roadmap annotations (for annotations derived from published scores) or the baseline-LD model, 8 Roadmap annotations, and corresponding published annotations (for annotations derived from boosted scores); (**c**) heritability enrichments of binary annotations derived from published and boosted baseline-LD model annotations; and (**d**) conditional $\tau^*$ values of binary annotations derived from published and boosted baseline-LD model annotations. In (**a**) and (**b**), we report results for the 10 most informative additional genome-wide scores (of 18 analyzed). In (**c**) and (**d**), we report results for the 10 most informative baseline-LD model annotations (of 47 analyzed). The percentage under each bar denotes the proportion of SNPs in the annotation; the proportion of top SNPs included in each annotation was optimized to maximize informativeness (largest $|\tau^*|$ among Bonferroni-significant annotations, or top 5% if no annotation was Bonferroni-significant; top 5% was the average optimized proportion among significant annotations). Error bars denote 95% confidence intervals. In panels (**b**) and (**d**), * denotes conditionally significant annotations. Numerical results are reported in Supplementary Data 14. Results for standardized enrichment, defined as enrichment times the standard deviation of annotation value (to adjust for annotation size), are reported in Supplementary Data 23.

baseline-LD model annotation. The CCR (Constrained Coding Regions) score[48] is a constraint score based on observed vs. expected variation in whole-exome sequence data; DeepSEA scores are described above. We note that the 18 additional scores included seven gene-based scores, which did not perform well; 3 published gene-based scores and four boosted gene-based scores

yielded conditionally significant binary annotations, but their $\tau^*$ were small (−0.02 to 0.09). Boosted versions of 3 of the remaining 5 additional scores and 20 of the remaining 23 baseline-LD model annotations had derived binary annotations that were significantly enriched for disease heritability (after Bonferroni correction) but not conditionally informative (Supplementary Data 14).

We performed six secondary analyses. First, for the 31 additional boosted scores which were conditionally significant and for which the underlying published scores had <100% of SNPs scored, we restricted the boosted scores to previously unscored variants and assessed the informativeness of the resulting binary annotations using S-LDSC. We determined that these annotations retained over half of the overall signals (average of 55% of absolute $\tau^*$; Supplementary Data 15), implying that AnnotBoost both imputes and denoises existing scores. Second, we again investigated which features of the baseline-LD model contributed the most to the informativeness of the boosted annotations by applying SHAP[41]. We determined that a broad set of features contributed to the predictions, including conservation-related features and LD-related features (as above), but also including regulatory features (e.g. H3K4me1, DGF, H3K9ac for boosted DeepSEA↑) (Supplementary Fig. 5). Third, we examined the trait-specific S-LDSC results, instead of meta-analyzing results across 41 traits. We identified nine (respectively 78) annotation-trait pairs with significant $\tau^*$ values for annotations derived from published (respectively boosted) scores (FDR < 5%; Supplementary Data 5). These annotation-trait pairs spanned five (respectively 32) different annotations, most of which (3/5 published, 23/32 boosted) were also conditionally significant in the meta-analysis across traits (Fig. 3b). Fourth, we assessed the heterogeneity of heritability enrichment and $\tau^*$ across the 41 traits (see "Methods"). We determined that 64/81 annotations tested had significant heterogeneity in enrichment and 40/81 annotations tested had significant heterogeneity in $\tau^*$, implying substantial heterogeneity across traits (Supplementary Data 6). Fifth, we repeated the analyses of Fig. 3 without including the eight Roadmap annotations. We determined that the number of significant binary annotations increased (Supplementary Data 16), confirming the importance of conditioning on the eight Roadmap annotations as an appropriate conservative step[39]. We further verified that including the eight Roadmap annotations did not impact results from previous sections (Supplementary Data 17). Sixth, we investigated the overlap between genes linked to each of our 43 conditionally significant annotations and 165 gene sets of biological importance (see "Methods"; Supplementary Data 7). As above, we consistently observed excess overlap for genes for which homozygous knockout in mice results in lethality[43,44] and high-pLI genes[38], and depleted overlap for olfactory receptor genes[45] (Supplementary Fig. 3, Supplementary Data 9 and 10), implying that our new annotations can help identify biologically important genes.

We conclude that six additional genome-wide annotations and 37 boosted annotations are conditionally informative for common disease, relative to baseline-LD model annotations.

**Constructing and evaluating combined heritability models**. We constructed and evaluated two heritability models incorporating our new functional annotations: (i) a combined marginal model incorporating all binary annotations that were conditionally significant (conditional on the baseline-LD model), and (ii) a combined joint model incorporating only a subset of binary annotations that were jointly and conditionally significant (conditional on each other and the baseline-LD model). We constructed the combined marginal model by merging the baseline-LD model with the 64 conditionally significant annotations (derived from 11 published scores and 53 boosted scores; Table 2) (baseline-LD + marginal). We constructed the combined joint model by performing forward stepwise elimination to iteratively remove annotations that had conditionally non-significant $\tau^*$ values after Bonferroni correction ($P \geq 0.05/500 = 0.0001$) or $\tau^* <$ 0.25 (see "Methods") (baseline-LD + joint). We have prioritized this type of combined joint model in the previous work[26,32,39,40,53], hypothesizing that there would be little value in retaining a large number of annotations containing redundant information; we note the substantial correlations between annotations in this study (Supplementary Data 18).

The combined joint model included 11 binary annotations derived from three published scores and eight boosted scores (Fig. 4, Table 2 and Supplementary Data 19). These 11 annotations are each substantially conditionally informative for common disease and include five boosted annotations with $|\tau^*| > 0.5$ (e.g., boosted ReMM: $\tau^* = 1.33$ (s.e. 0.12)); annotations with $|\tau^*| > 0.5$ are unusual, and considered to be very important[40]. We note that the top 0.5% of SNPs from REVEL↑[6] had significantly negative $\tau^*$ (−0.95 (s.e. 0.08)), as the annotation was significantly enriched for disease heritability but less enriched than expected based on annotations from the combined joint model; as noted above, annotations with significantly positive or negative $\tau^*$ are conditionally informative.

We performed two analyses to evaluate the combined joint model and the combined marginal model, compared to the baseline-LD model; these analyses evaluated aggregate heritability models, rather than individual functional annotations. First, we computed the average $\log l_{SS}$[57] (an approximate likelihood metric) of each model, relative to a model with no functional annotations, across 30 common diseases and complex traits from the UK Biobank[58] (subset of 41 traits; Supplementary Table 1) ($\Delta \log l_{SS}$; see Methods). Results are reported in Fig. 5a and Supplementary Data 20. The combined joint model attained a +7.4% larger $\Delta \log l_{SS}$ than the baseline-LD model ($P < 6e−38$); the combined marginal model attained a +23.9% larger $\Delta \log l_{SS}$ than the baseline-LD model ($P < 2e−99$), including a significantly larger improvement for 30/30 traits analyzed (Fig. 5b and Supplementary Data 20). The improvements were only slightly smaller when using Akaike Information Criterion (AIC) to account for increases in model complexity[57] (+7.0% and +20.3%; Supplementary Data 20).

Second, we assessed each model's accuracy of classifying three different sets of fine-mapped SNPs (from 10 LD-, MAF-, and genomic-element-matched control SNPs in the reference panel[28]): 7,333 fine-mapped for 21 autoimmune diseases from Farh et al.[59], 3768 fine-mapped SNPs for inflammatory bowel disease from Huang et al.[60], and 1851 fine-mapped SNPs for 49 UK Biobank traits from Weissbrod et al.[61]. We note that with the exception of Weissbrod et al. fine-mapped SNPs (stringently defined by causal posterior probability ≥ 0.95; FDR < 0.05), 95% credible fine-mapped SNPs likely include a large fraction of non-causal variants. We computed the AUPRC attained by the combined joint model and the combined marginal model, relative to a model with no functional annotations (ΔAUPRC), aggregated by training a gradient boosting model (multi-score analysis); we used odd (respectively even) chromosomes as training data to make predictions for even (respectively odd) chromosomes (see "Methods"). We note that this gradient boosting model uses disease data (fine-mapped SNPs), whereas AnnotBoost does not use disease data to construct boosted pathogenicity scores; specifically, our boosted scores do not use fine-mapped SNPs. The combined joint model attained a +2.5% to 6.9% larger ΔAUPRC than the baseline-LD model (each $P < 3e−28$); the combined marginal model attained a +4.9% to 21.3% larger ΔAUPRC than the baseline-LD model (each $P < 7e−100$); we obtained similar results using AUROC (Supplementary Fig. 6, Supplementary Data 21). This improvement likely comes from non-linear interactions involving the boosted annotations, published annotations, and the baseline-LD model.

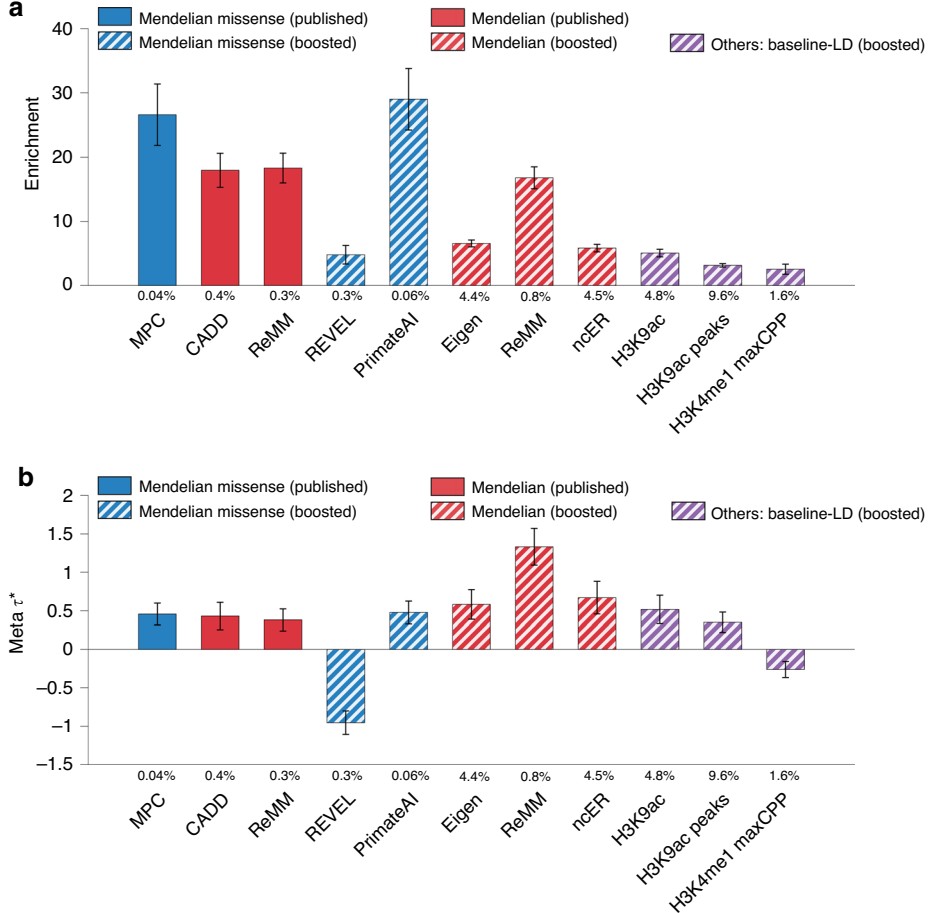

**Fig. 4 Informativeness for a common disease of 11 jointly significant binary annotations from combined joint model.** We report (**a**) heritability enrichment of 11 jointly significant binary annotations, meta-analyzed across 41 independent traits; **b** joint $\tau^*$ values, conditioned on the baseline-LD model, eight Roadmap annotations, and each other. We report results for the 11 jointly conditionally informative annotations in the combined joint model (S-LDSC $\tau^* P < 0.0001$ and $|\tau^*| \geq 0.25$). The percentage under each bar denotes the proportion of SNPs in the annotation. Error bars denote 95% confidence intervals. Numerical results are reported in Supplementary Data 19. Results for standardized enrichment, defined as enrichment times the standard deviation of annotation value (to adjust for annotation size), are reported in Supplementary Data 23.

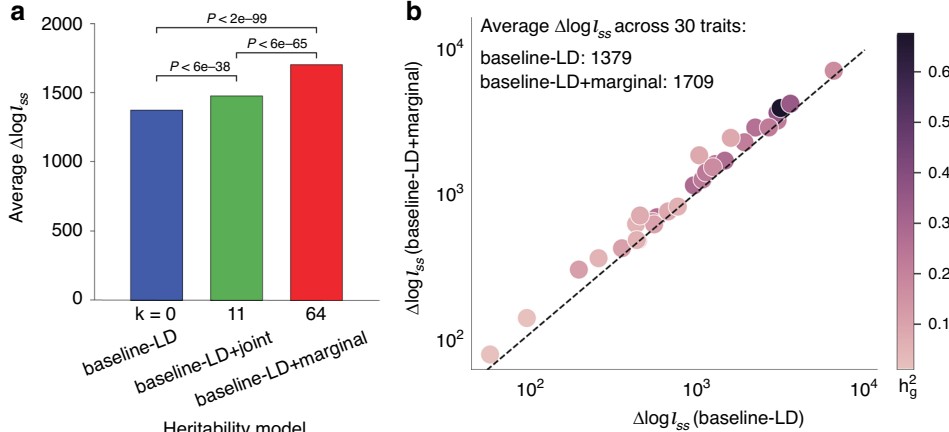

**Fig. 5 Evaluation of improvement in heritability model fit.** We report (**a**) average $\Delta\log l_{SS}$ (an approximate model likelihood metric[57]) across 30 UKBB traits; **b** $\Delta\log l_{SS}$ of the baseline-LD and baseline-LD + marginal models for each trait. $\Delta\log l_{SS}$ is computed as $\log l_{SS}$ of a given model - $\log l_{SS}$ of a model with no functional annotations (baseline-LD-nofunct model: MAF/LD annotations only). In (**a**), $k$ denotes the number of new annotations beyond the baseline-LD model. Numerical results are reported in Supplementary Data 20.

We performed eight secondary analyses. First, we repeated $\log l_{SS}$ analysis on the model with 19 new annotations with conditional $\tau^* > 0.5$; we determined this model attained a +10.6% larger $\Delta \log l_{SS}$ and +9.7% larger AIC than the baseline-LD model ($P < 2e-50$) (Supplementary Data 20). Second, we applied SHAP[41] to investigate which features contributed the most to classification of fine-mapped SNPs; we determined that boosted scores often drove the predictions, validating the potential utility of boosted scores in functionally informed fine-mapping (e.g., H3K9ac↑, CADD) (Supplementary Figs. 7 and 8). Third, we repeated the classification of fine-mapped SNPs using a single LD-, MAF-, and genomic-element-matched control variant (instead of 10 control variants) for each fine-mapped SNP, and obtained similar results (Supplementary Data 21). Fourth, we repeated the classification of fine-mapped disease SNPs analysis of Weissbrod et al. fine-mapped SNPs using 1379 SNPs that were fine-mapped without using functional information[61] (to ensure that results were not circular), and obtained similar results (Supplementary Fig. 6, Supplementary Data 21). Fifth, we computed the AUPRCs for classifying fine-mapped SNPs individually attained by each of 82 published and 82 boosted scores (single-score analysis), comparing results for boosted scores vs. the corresponding published scores. The boosted scores significantly outperformed the corresponding published scores in each case (66/82 to 80/82 scores; Supplementary Fig. 9, Supplementary Data 22 and 23). We also found that AUPRC and AUROC results for published and boosted scores were moderately correlated with S-LDSC results (up to $r = 0.67$) for binary annotations derived from these scores, validating the S-LDSC results (Supplementary Data 24). Sixth, we repeated the single-score and multi-score analysis using 14,807 NHGRI GWAS SNPs[62,63]; we obtained similar results (Supplementary Figs. 6 and 9, Supplementary Data 21 and 22). Seventh, we computed genome-wide correlations between all annotations analyzed including baseline-LD model annotations (Supplementary Data 18). Several of the jointly significant annotations were strongly correlated (up to 0.73) with conservation-related annotations from the baseline-LD model, particularly binary GERP scores, consistent with our SHAP results (Supplementary Figs. 2, 4, and 5). Eighth, we compared the informativeness of the baseline-LD model and the combined joint model. We identified the addition of 11 jointly significant annotations greatly reduced the informativeness of several existing baseline-LD annotations, including conservation-related annotations (e.g., conserved primate, binary GERP scores) and other annotations (e.g., coding, CpG content; see Supplementary Fig. 10 and Supplementary Data 25), recapitulating the informativeness of 11 jointly significant annotations.

We conclude that the combined joint model and the combined marginal model both significantly outperformed the baseline-LD model, validating the informativeness of our new annotations for common disease. The improvement was much larger for the combined marginal model; this finding was surprising, in light of our previous work advocating for conservatively restricting to jointly significant annotations when expanding heritability models[26,32,39,40,53]. However, we caution that due to the much larger number of new annotations in the combined marginal model, the combined joint model may still be preferred in some settings.

## Discussion
We analyzed the informativeness of a broad set of Mendelian disease-derived pathogenicity scores across 41 independent common diseases and complex traits to show that several annotations derived from published Mendelian disease-derived scores

were conditionally informative for the common disease after conditioning on the baseline-LD model. We further developed AnnotBoost, a gradient boosting-based machine-learning framework to impute and denoise existing pathogenicty scores. We determined that annotations derived from boosted pathogenicity scores were even more informative for common disease, resulting in 64 marginally significant annotations and 11 jointly significant annotations and implying that Mendelian disease variants and common disease variants share similar properties. These variant-level results are substantially different from previous studies of gene-level overlap between Mendelian diseases and complex traits[12–19]. Notably, our new annotations produced significant improvements in heritability model fit and in classifying disease-associated, fine-mapped SNPs. We also detected significant excess overlap between genes linked to our new annotations and biologically important gene sets.

We note three key differences between AnnotBoost and previous approaches that utilized gradient boosting to identify pathogenic missense[7] and non-coding variants[9,10]. First, AnnotBoost uses a pathogenicity score as the only input and does not use disease data (e.g., ClinVar[29] or HGMD[30]). Second, AnnotBoost produces genome-wide scores, even when some SNPs are unscored by the input pathogenicity score. Third, AnnotBoost leverages 75 diverse features from the baseline-LD model[26,27], significantly more than the previous approaches[7,9,10]. Indeed, we determined that AnnotBoost produces strong signals even when conditioned on those approaches.

Our findings have several ramifications for improving our understanding of the common disease. First, elucidating specific mechanistic links between Mendelian disease and common disease may yield important biological insights. Second, it is of interest to assess the informativeness for the common disease of Mendelian disease pathogenicity scores that may be developed in the future, particularly after imputing and denoising these scores using AnnotBoost; this would further elucidate the shared properties between Mendelian disease variants and common disease variants. Third, annotations derived from published and boosted Mendelian pathogenicity scores can be used to improve functionally informed fine-mapping[61,64–67], motivating their inclusion in future large-scale fine-mapping studies. (On the other hand, we anticipate that our new annotations will be less useful for improving functionally informed polygenic risk prediction[68,69] and association mapping[70], because there is pervasive LD between SNPs in an annotation and SNPs outside of an annotation, such that these annotations do not distinguish which LD blocks contain causal signal.) Fourth, the larger improvement for our combined marginal model versus our combined joint model (Fig. 5a) advocates for a more inclusive approach to expanding heritability models, as compared to our previous work advocating for conservatively restricting to jointly significant annotations[26,32,39,40,53]. However, the combined marginal model suffers a cost of reduced interpretability (it contains a much larger number of new annotations, and it is unclear which of these annotations are providing the improvement), thus the combined joint model may still be preferred in some settings. Fifth, gene scores derived from published and boosted Mendelian pathogenicity scores can be used to help identify biologically important genes; we constructed gene scores by linking SNPs to their nearest gene, but better strategies for linking regulatory variants to genes[71–73] could potentially improve upon our results.

We note several limitations of our work. First, we focused our analyses on the common disease (which are driven by common and low-frequency variants) and did not analyze Mendelian diseases (which are driven by very rare variants); the application of AnnotBoost to impute and denoise very rare pathogenic variants for Mendelian disease is a direction for future work. Second,

we primarily report results that are meta-analyzed across 41 traits (analogous to the previous studies[25,26,32,39,40,53]), but results and their interpretation may vary substantially across traits. Nonetheless, our combined marginal model produced a significant improvement in heritability model fit for 30/30 UK Biobank traits analyzed (Fig. 5b). Third, S-LDSC is not well-suited to the analysis of annotations spanning a very small proportion of the genome, preventing the analysis of a subset of published pathogenicity scores; nonetheless, our main results attained high statistical significance. Fourth, we restricted all of our analyses to European populations, which have the largest available GWAS sample size. However, we expect our results to be generalizable to other populations, as functional enrichments have been shown to be highly consistent across ancestries[65,74,75]; we note that assessing functional enrichments in admixed populations[76] would require the application of an unpublished extension of S-LDSC[77]. Fifth, the gene-based SNP scores that we analyzed did not perform well, perhaps because they were defined using 100kb windows, a crude strategy employed in the previous work[32,53,78]; better strategies for linking regulatory variants to genes[71–73] (as shown in above gene scores) could potentially improve upon those results. Despite these limitations, the imputed and denoised pathogenicity scores produced by our AnnotBoost framework have high potential to improve gene discovery and fine-mapping for common disease.

## Methods

**Genomic annotations and the baseline-LD model.** We define a genomic annotation as an assignment of a numeric value to each SNP above a specified minor allele frequency (e.g., MAF ≥ 0.5%) in a predefined reference panel (e.g., 1000 Genomes[28]). Continuous-valued annotations can have any real value. Probabilistic annotations can have any real value between 0 and 1. A binary annotation can be viewed as a subset of SNPs (the set of SNPs with annotation value 1); we note all annotations analyzed in this work are binary annotations. Annotations that correspond to known or predicted functions are referred to as functional annotations.

The baseline-LD model[26] (v2.1) contains 86 functional annotations (see "Data availability"). We use these annotations as features of AnnotBoost (see below). These annotations include genomic elements (e.g., coding, enhancer, promoter), conservation (e.g., GERP, PhastCon), regulatory elements (e.g., histone marks, DNaseI-hypersensitive sites (DHS), transcription factor (TF) binding sites), and LD-related annotations (e.g., predicted allele age, recombination rate, SNPs with low levels of LD).

**Enrichment and $\tau^*$ metrics.** We used stratified LD score regression (S-LDSC[25,26]) to assess the contribution of an annotation to disease heritability by estimating the enrichment and the standardized effect size ($\tau^*$) of an annotation.

Let $a_{cj}$ represent the (binary or probabilistic) annotation value of the SNP $j$ for the annotation $c$. S-LDSC assumes the variance of per normalized genotype effect sizes is a linear additive contribution to the annotation $c$:

$$\text{Var}(\beta_j) = \sum_c a_{cj}\tau_c \tag{1}$$

where $\tau_c$ is the per-SNP contribution of the annotation $c$. S-LDSC estimates $\tau_c$ using the following equation:

$$\text{E}[\chi_j^2] = N\sum_c \ell(j, c)\tau_c + 1 \tag{2}$$

where $N$ is the sample size of the GWAS and $\ell(j, c)$ is the LD score of the SNP $j$ to the annotation $c$. The LD score is computed as follow $\ell(j, c) = \sum_k a_{ck}r_{jk}^2$ where $r_{jk}$ is the correlation between the SNPs $j$ and $k$.

We used two metrics to assess the informativeness of an annotation. First, the standardized effect size ($\tau^*$), the proportionate change in per-SNP heritability associated with a one standard deviation increase in the value of the annotation (conditional on all the other annotations in the model), is defined as follows:

$$\tau_c^* = \frac{\tau_c sd(C)}{h_g^2/M} \tag{3}$$

where $sd(C)$ is the standard deviation of the annotation $c$, $h_g^2$ is the estimated SNP-heritability, and $M$ is the number of variants used to compute $h_g^2$ (in our experiment, $M$ is equal to 5,961,159, the number of common SNPs in the reference panel). The significance for the effect size for each annotation, as mentioned in the previous studies[26,32,53], is computed as ($\frac{\tau^*}{se(\tau^*)} \sim N(0, 1)$), assuming that $\frac{\tau^*}{se(\tau^*)}$ follows a normal distribution with zero mean and unit variance.

Second, enrichment of the binary and probabilistic annotation is the fraction of heritability explained by SNPs in the annotation divided by the proportion of SNPs in the annotation, as shown below:

$$\text{Enrichment} = \frac{\%h_g^2(C)}{\%\,\text{SNP}\,(C)} = \frac{\frac{h_g^2(C)}{h_g^2}}{\frac{\sum_j a_{jc}}{M}} \tag{4}$$

where $h_g^2(C)$ is the heritability captured by the $c$th annotation. When the annotation is enriched for trait heritability, the enrichment is >1; the overlap is greater than one would expect given the trait heritablity and the size of the annotation. The significance for enrichment is computed using the block jackknife as mentioned in the previous studies[25,32,53,78]. The key difference between enrichment and $\tau^*$ is that $\tau^*$ quantifies effects that are unique to the focal annotation after conditioning on all the other annotations in the model, while enrichment quantifies effects that are unique and/or non-unique to the focal annotation.

In all our analyses, we used the European samples in 1000 G[28] (see "Data availability") as reference SNPs. Regression SNPs were obtained from HapMap 3[79] (see "Data availability"). SNPs with marginal association statistics >80 and SNPs in the major histocompatibility complex (MHC) region were excluded. Unless stated otherwise, we included the baseline-LD model[26] in all primary analyses using S-LDSC, both to minimize the risk of bias in enrichment estimates due to model mis-specification[25,26] and to estimate effect sizes ($\tau^*$) conditional on known functional annotations.

**Published Mendelian disease-derived pathogenicity scores.** We considered a total 35 published scores: 11 Mendelian disease-derived missense pathogenicity scores, 6 genome-wide Mendelian disease-derived pathogenicity scores, and 18 additional scores (see Table 1 and Supplementary Data 13). Here, we provide a short description for Mendelian missense and genome-wide Mendelian disease-derived pathogenicity scores. Details for 18 additional scores and the baseline-LD annotations are provided in Supplementary Data 13. Our curated pathogenicity scores are available online (see "Data availability").

For all scores, we constructed annotations using GRCh37 (hg19) assembly limited to all 9,997,231 low-frequency and common SNPs (with MAF ≥ 0.5%) found in 1000 Genomes[28] European Phase 3 reference genome individuals. Mendelian missense scores were readily available from dbNSFP database[80,81] using a rank score (a converted score based on the rank among scored SNPs); genome-wide Mendelian disease-derived scores were individually downloaded and used with no modification to original scores (see "Data availability"). For each pathogenicity score, we constructed a binary annotation based on an optimized threshold (See below). Short descriptions for each pathogenicity score (excluding 18 additional scores and the baseline-LD annotations; provided in Supplementary Data 13) are provided below:

Mendelian disease-derived missense pathogenicity scores:

*PolyPhen-2*[1,33]*(HDIV and HVAR)*: Higher scores indicate a higher probability of the missense mutation being damaging on the protein function and structure. The default predictor is based on a naive Bayes classifier using HumDiv (HDIV), and the other is trained using HumVar (HVAR), using eight sequence-based and three structure-based features.

*MetaLR/MetaSVM*[34]: An ensemble prediction score based on logistic regression (LR) or support vector machine (SVM) to classify pathogenic mutations from background SNPs in whole-exome sequencing, combining nine prediction scores and one additional feature (maximum minor allele frequency).

*PROVEAN*[35,82]: An alignment-based score to predict the damaging single amino acid substitutions.

*SIFT 4G*[5]: Predicted deleterious effects of an amino acid substitution to protein function based on sequence homology and physical properties of amino acids.

*REVEL*[6]: An ensemble prediction score based on a random forest classifier trained on 6182 missense disease mutations from HGMD[30], using 18 pathogenicity scores as features.

*M-CAP*[7]: An ensemble prediction score based on a gradient boosting classifier trained on pathogneic variants from HGMD[30] and benign variants from ExAC data set[38], using nine existing pathogenicity scores, seven base-pair, amino acid, genomic region, and gene-based features, and four features from multiple sequence alignments across 99 species.

*PrimateAI*[8]: A deep-learning-based score trained on the amino acid sequence flanking the variant of interest and the orthologous sequence alignments in other species and eliminating common missense variants identified in six non-human primate species.

*MPC*[36]*(missense badness, PolyPhen-2, and constraint)*: Logistic regression-based score to identify regions within genes that are depleted for missense variants in ExAC data[38] and incorporating variant-level metrics to predict the impact of missense variants. Higher MPC score indicates increased deleteriousness of amino acid substitutions once occurred in missense-constrained regions.

*MVP*[37]: A deep-learning-based score trained on 32,074 pathogenic variants from ClinVar[29], HGMD[30], and UniProt[83], using 38 local contexts, constraint,

conservation, protein structure, gene-based, and existing pathogenicity scores as features.

Genome-wide Mendelian disease-derived pathogenicity scores:

*CADD*[2,46]: An ensemble prediction score based on a support vector machine classifier trained to differentiate 14.7 million high-frequency human-derived alleles from 14.7 million simulated variants, using 63 conservation, regulatory, protein-level, and existing pathogenicity scores as features. We used PHRED-scaled CADD score for all possible SNVs of GRCh37.

*Eigen/Eigen-PC*[3]: Unsupervised machine-learning score based on 29 functional annotations and leveraging blockwise conditional independence between annotations to differentiate functional vs. non-functional variants. Eigen-PC uses the lead eigenvector of the annotation covariance matrix to weight the annotations. For both Eigen and Eigen-PC, we used PHRED-scaled scores and combined coding and non-coding regions to make it as a single genome-wide score. Higher score indicates more important (predicted) functional roles.

*ReMM*[4]*(regulatory Mendelian mutation)*: An ensemble prediction score based on a random forest classifier to distinguish 406 hand-curated Mendelian mutations from neutral variants using conservation scores and functional annotations. Higher ReMM score indicates the greater potential to cause a Mendelian disease if mutated.

*NCBoost*[10]: An ensemble prediction score based on a gradient boosting classifier trained on 283 pathogenic non-coding SNPs associated with Mendelian disease genes and 2830 common SNPs, using 53 conservation, natural selection, gene-based, sequence context, and epigenetic features.

*ncER*[9]*(non-coding essential regulation)*: An ensemble prediction score based on a gradient boosting classifier trained on 782 non-coding pathogenic variants from ClinVar[29] and HGMD[30], using 38 gene essentiality, 3D chromatin structure, regulatory, and existing pathogenicity scores as features.

**AnnotBoost framework**. AnnotBoost is based on gradient boosting, a machine-learning method for classification; the AnnotBoost model is trained using the XGBoost gradient boosting software[31] (see "Code availability"). AnnotBoost requires only one input, a pathogenicity score to boost, and generates a genome-wide (probabilistic) pathogenicity score (as described in Supplementary Fig. 1). During the training, AnnotBoost uses decision trees, where each node in a tree splits SNPs into two classes (pathogenic and benign) using 75 codings, conserved, regulatory, and LD-related features from the baseline-LD model[26] (excluding 10 MAF bins features; we obtained similar results with or without MAF bins features; see Supplementary Fig. 11). We note that the baseline-LD annotations considered all low-frequency and common SNPs thus do not have unscored SNPs. The method generates training data from the input pathogenicity scores without using external variant data; top 10% SNPs from the input pathogenicity score are labeled as a positive training set, and the bottom 40% SNPs are labeled as a control training set; we obtained similar results with other training data ratios (see Supplementary Fig. 12). The prediction is based on $T$ additive estimators (we use $T = 200$–$300$; see below), minimizing the following loss objective function $L^t$ at the $t$-th iteration:

$$L^t = \sum_{i=1}^{n} l(y_i, \hat{y}_i^{t-1} + f_t(x_i)) + \gamma(f_t) \qquad (5)$$

where $l$ is a differentiable convex loss function (which measures the difference between the prediction ($\hat{y}_i$) and the target $y_i$ at the $i$-th instance), $f_t$ is an independent tree structure, and last term $\gamma(f_t)$ penalizes the complexity of the model, helping to avoid over-fitting. The prediction ($\hat{y}_i$) is made by $\sum_{t=1}^{T} f_t(x_i)$ by ensembling outputs of multiple weak-learner trees. Odd (respectively even) chromosome SNPs are used for training to score even (respectively odd) chromosome SNPs. The output of the classifier is the probability of being similar to the positive training SNPs and dissimilar to the control training SNPs.

We used the following model parameters: the number of estimators (200, 250, 300), depth of the tree (25, 30, 35), learning rate (0.05), gamma (minimum loss reduction required before additional partitioning on a leaf node; 10), minimum child weight (6, 8, 10), and subsample (0.6, 0.8, 1); we optimized parameters with hyperparamters tuning (a randomized search) with fivefold cross-validation. Two important parameters to avoid over-fitting are gamma and learning rate; we chose these values consistent with the previous studies[9,10]. The model with the highest AUROCs on the held-out data was selected and used to make a prediction.

To identify which feature(s) drives the prediction output with less bias, AnnotBoost uses Shapley Additive Explanation (SHAP)[41], a widely used tool to interpret complex non-linear models, instead of built-in feature importance tool because of SHAP's property of satisfying symmetry, dummy player, and additivity axioms. SHAP uses the training matrix (features × SNP labels) and the trained model to generate a signed impact of each baseline-LD features on the AnnotBoost prediction.

To evaluate the performance of classifiers, we plotted receiver operating characteristic (ROC) and precision-recall (PR) curves. As we train AnnotBoost by splitting SNPs into odd and even chromosomes, we report the average out-of-sample area under the curve (AUC) of the odd and even chromosomes classifier. We used the threshold of 0.5 to define a class; that is, class 1 includes SNPs with the output probability > 0.5. We caution that high classification accuracy does not necessarily translate into conditional informativeness for common disease[39].

**Constructing binary annotations using top variants from published and boosted scores**. For published Mendelian disease-derived missense pathogenicity scores, we considered five different thresholds to construct binary annotations: top 50, 40, 30, 20 or 10% of scored variants. For published scores that produce Bonferroni-significant binary annotations, we report results for the binary annotation with the largest $|\tau^*|$ among those that are Bonferroni-significant. For published scores that do not produce Bonferroni-significant binary annotations, we report results for the threshold with the most significant $\tau^*$ (even though not Bonferroni-significant).

For all other published pathogenicity scores, we considered the top 10, 5, 1, 0.5 or 0.1% of scored variants to construct binary annotations; we used more inclusive thresholds for published Mendelian disease-derived missense pathogenicity scores due to the small proportion of variants scored (~0.3%; see Table 1). For published scores that produce Bonferroni-significant binary annotations, we report results for the binary annotation with the largest $|\tau^*|$ among those that are Bonferroni-significant. For published scores that do not produce Bonferroni-significant binary annotations, we report results for the top 5% of variants (the average optimized proportion among Bonferroni-significant binary annotations); we made this choice because (in contrast to published Mendelian missense scores) for many other published scores the most significant $\tau^*$ was not even weakly significant.

For boosted pathogenicity scores, we considered the top 10, 5, 1, 0.5 or 0.1% of scored variants, as well as variants with boosted scores ≥0.5; we note that top 10% of SNPs does not necessarily translate to 10% of SNPs, as some SNPs share the same score, and some genomic regions (e.g., MHC) are excluded when running S-LDSC (see below). For boosted scores that produce Bonferroni-significant binary annotations, we report results for the binary annotation with largest $|\tau^*|$ among those that are Bonferroni-significant. For boosted scores that do not produce Bonferroni-significant binary annotations, we report results for the top 5% of variants.

In all analyses, we excluded binary annotations with a proportion of SNPs <0.02% (the same threshold used in ref. [53]), because S-LDSC does not perform well for small annotations[25]. We analyzed 155 annotations derived from published scores (31 published scores (Table 2), 5 thresholds for top x% of variants, $31 \times 5 = 155$), such that 500 hypotheses is a conservative correction in the analysis of published scores. We also analyzed 492 annotations derived from boosted scores (82 underlying published scores including 47 baseline-LD model annotations (Table 2), 6 thresholds for top x% of variants, $82 \times 6 = 492$), such that 500 hypotheses is a roughly appropriate correction in the analysis of boosted scores. For simplicity, we corrected for $\max(155,492) \approx 500$ hypotheses throughout. We note that, in the meta-analysis $\tau^*$ p-values, a global FDR < 5% corresponds to $P < 0.0305$; thus, our choice of $P < 0.05/500 = 0.0001$ is conservative.

In all primary analyses, we analyzed only binary annotations. However, we verified in a secondary analysis of the CDTS score[47] that probabilistic annotations produced results similar to binary annotations (see Supplementary Fig. 13).

**Heterogeneity of enrichment and $\tau^*$ across traits**. For a given annotation, we assessed the heterogeneity of enrichment and $\tau^*$ (across 41 independent traits) by estimating the standard deviation of the true parameter value across traits, as analogous to ref. [23]. We calculated the cross-trait $\tau^*$ as the inverse variance weighted mean across the traits. Then, we compared $\sum_{i=1}^{n} (\hat{\tau}_i - \hat{\tau}_{\text{across}-\text{trait}})^2 / (\text{std.error}_i^2)$ to a $\chi_n^2$ null statistic, where $n = 47$ (41 independent traits; 47 summary statistics; see Supplementary Table 1). We repeated the analysis for heritability enrichment by using enrichments and standard errors of enrichment estimates from S-LDSC.

**Overlap between gene score quintiles informed by input pathogenicity scores and 165 reference gene sets**. For a given pathogenicity score, we scored genes based on the maximum pathogenicity score of linked SNPs, where SNPs were linked to a unique nearest gene using ANNOVAR[84]: 9,997,231 SNP-gene links, decreasing to 5,059,740 S2G links after restricting to 18,117 genes with a protein product (according to HGNC[85]) that have an Ensembl gene identifier (ENSG ID). Gene scores are reported in Supplementary Data 7. We constructed quintiles of gene scores and assessed gene-level excess fold overlap with 165 reference gene sets of biological importance (see below; summarized in Supplementary Data 8). We note that this analysis used continuous-valued pathogenicity scores, instead of binary annotations.

The 165 reference gene sets (Supplementary Data 8) reflected a broad range of gene essentiality[86] metrics, as outlined in ref. [53]. They included known phenotype-specific Mendelian disease genes[19], constrained genes[38,87–89], essential genes[43,44,90], specifically expressed genes across GTEx tissues[78], dosage outlier genes across GTEx tissues[91], genes with a ClinVar pathogenic or likely pathogenic variants[29], genes in the Online Mendelian Inheritance in Man (OMIM[92]), high network connectivity genes in different gene networks[53,93], genes with more independent SNPs[53], known drug targets[94], human targets of FDA-approved drugs[95], eQTL-deficient genes[96,97], and housekeeping genes[98]; a subset of these gene sets were previously analyzed in ref. [53].

As defined in our previous study[53], the excess fold overlap of gene set 1 and gene set 2 is defined as follows:

$$\text{excessoverlap(geneset1, geneset2)} = P_d/P_{tot} \qquad (6)$$

where $P_d = \frac{|\text{geneset1} \cap \text{geneset2}|}{|\text{geneset2}|}$ and $P_{tot} = \frac{|\text{geneset1} \cap \text{allprotein} - \text{codinggenes}|}{|\text{allprotein} - \text{codinggenes}|}$. The standard error for the excess overlap is similarly scaled:

$$SE = \sqrt{\frac{P_d(1 - P_d)}{|\text{geneset2}|}} / P_{tot} \qquad (7)$$

When there is excess overlap, the excess fold overlap is >1; when there is depletion, the excess fold overlap is <1. We assessed the odds ratio and significance in the difference between the excess overlap between the boosted gene quintile and the published gene quintile by the Fisher's exact test (two-sided).

**Evaluating heritability model fit using** $\log l_{SS}$. Given a heritability model (e.g., the baseline-LD model, combined joint model, or combined marginal model), we define the $\Delta \log l_{SS}$ of that heritability model as the $\log l_{SS}$ of that heritability model minus the $\log l_{SS}$ of a model with no functional annotations (baseline-LD-nofunct; 17 LD and MAF annotations from the baseline-LD model[26]), where $\log l_{SS}$[57] is an approximate likelihood metric that has been shown to be consistent with the exact likelihood from restricted maximum likelihood (REML; see Code availability). We compute p-values for $\Delta \log l_{SS}$ using the asymptotic distribution of the Likelihood Ratio Test (LRT) statistic: $-2\log l_{SS}$ follows a $\chi^2$ distribution with degrees of freedom equal to the number of annotations in the focal model, so that $-2\Delta \log l_{SS}$ follows a $\chi^2$ distribution with degrees of freedom equal to the difference in number of annotations between the focal model and the baseline-LD-nofunct model. We used UK10K as the LD reference panel and analyzed 4,631,901 HRC (haplotype reference panel[99]) well-imputed SNPs with MAF ≥0.01 and INFO ≥ 0.99 in the reference panel; We removed SNPs in the MHC region, SNPs explaining >1% of phenotypic variance and SNPs in LD with these SNPs.

We computed $\Delta \log l_{SS}$ for four heritability models:

- baseline-LD: annotations from the baseline-LD model[25,26] (86 annotations)
- baseline-LD + joint: baseline-LD model + 11 jointly significant annotations (3 published, 8 boosted; 97 annotations)
- baseline-LD + marginal-stringent: baseline-LD model + 19 marginally significant annotations with conditional $|\tau^*| > 0.5$ (105 annotations)
- baseline-LD + marginal: baseline-LD model + 64 marginally significant annotations (11 published, 53 boosted; 150 annotations)

**Classification of fine-mapped disease SNPs**. We assessed the classification accuracy of fine-mapped disease SNPs. Here, we consider only low-frequency and common SNPs (MAF ≥ 0.5%) and report the total number of unique SNPs (regardless MAF). we assessed the accuracy of classifying five different SNP sets (summarized in Supplementary Data 22): (1) 7333 fine-mapped for 21 autoimmune diseases from Farh et al.[59] (of 7747 total SNPs; 95% credible sets), (2) 3768 fine-mapped SNPs for inflammatory bowel disease from Huang et al.[60] (of 4311 total SNPs; 95% credible sets), (3) 1851 SNPs (of 2225 SNPs, spanning 3025 SNP-trait pairs; stringently defined by causal posterior probability ≥ 0.95) functionally informed fine-mapped for 49 UK Biobank traits from Weissbrod et al.[61], (4) 1379 (of 1853 total SNPs with causal posterior probability ≥ 0.95) non-functionally informed fine-mapped SNPs for 49 UK Biobank traits from Weissbrod et al.[61], and (5) 14,807 SNPs from the NHGRI GWAS catalog[62,63] (2019-07-12 version; p-value < 5e−8; we note only about 5% of GWAS SNPs are expected to be causal[59]).

For each of these five SNP sets, we matched 10 control SNPs for each positive fine-mapped SNP by matching LD, MAF, and genomic element, as in the previous studies[9,10,47]; we note that these studies emphasized the need for matching the relative genomic region distribution in performance evaluation. MAF was based on the same reference panel (European samples from 1000 Genomes Phase 3[28]), and LD was estimated by applying S-LDSC on all SNPs annotation ('base'). To identify the genomic element of each SNPs and the nearest gene, we annotated these five sets of SNPs using ANNOVAR[84] using the gene-based annotation. For assigning the genomic element to each SNP, we used the default ANNOVAR prioritization rule for gene-based annotations: exonic = splicing (defined by 10bp of a splicing junction) > ncRNA > UTR5 = UTR3 > intronic > upstream = downstream > intergenic. When SNP (in the intergenic or intronic region) is associated with overlapping genes, the nearest protein-coding gene (based on the distance to the TSS or TSE) is retained. To obtain 10 control SNPs for each positive fine-mapped SNP, we first searched the control SNPs within the same genomic element and the same chromosome of that positive SNP; then kept the 10 control SNPs with the most similar LD and MAF (based on the average of rank(LD difference from the positive SNP) and rank(MAF difference from the positive SNP)). In secondary analyses, we instead retained a unique most closely matched control SNP.

Given positive and control SNP sets, we computed the AUPRCs (and AUROCs) by an individual score (each of 82 published and 82 boosted scores). We refer this as a single-score analysis. We used AUPRC as a primary metric, as AUPRC is more robust for imbalanced data[100]. We assessed the significance of the difference between two AUPRCs using 1000 samples bootstrapped standard errors then performed two-sample $t$-test; variance of AUPRCs (and AUROCs) from 1000 samples was sufficiently small. We note this single-score analysis measures an improvement between two scores, where one score is derived from the other (e.g., our boosted gene from published score). Also, we performed a multi-score

analysis. For each heritability model, we aggregated scores by training a gradient boosting model (features: aggregated scores, positive label: each of five sets of SNPs, control label: LD-, MAF-, and genomic-element-matched control sets of SNPs); we used odd (respectively even) chromosomes as training data to make predictions for even (respectively odd) chromosomes. We used the same training parameters as AnnotBoost (carefully selected to avoid over-fitting, consistent with the previous study[9,10]) with hyperparameters tuned using a randomized search method with fivefold cross-validation. We report the average AUPRC and AUROC of odd and even chromosome classifiers. We also computed ΔAUPRC as AUPRC of a given model minus AUPRC of baseline-LD-nofunct model. We note that no disease data (five sets of SNPs used as labels) was re-used in these analyses, as AnnotBoost uses only the input pathogenicity scores to generate positive and negative sets of training data. We assessed the significance of the difference as described above. To identify which feature(s) drives the prediction output, we applied SHAP[41] to generate a signed impact of each baseline-LD, published, and boosted score features on classifying fine-mapped disease SNPs.

**Reporting summary**. Further information on research design is available in the Nature Research Reporting Summary linked to this article.

## Data availability
All published and boosted pathogenicity scores and binary annotations and SHAP results are available at https://alkesgroup.broadinstitute.org/LDSCORE/Kim_annotboost/. GWAS summary statistics are available at https://alkesgroup.broadinstitute.org/LDSCORE/independent_sumstats/. The baseline-LD annotations (v.2.1) are available at https://alkesgroup.broadinstitute.org/LDSCORE/. The 1000 Genomes Project Phase 3 data are available at ftp://ftp.1000genomes.ebi.ac.uk/vol1/ftp/release/20130502. 165 reference gene sets are available at https://github.com/samskim/networkconnectivity.

## Code availability
AnnotBoost source code is provided here: https://github.com/samskim/annotboost/. This work primarily uses the S-LDSC software (https://github.com/bulik/ldsc). SumHer software for computing $\log l_{SS}$ is available at http://dougspeed.com/sumher/.

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

## Acknowledgements

We are grateful to Bryce van de Geijn, Farhad Hormozdiari, Armin Schoech, and Huwenbo Shi for helpful discussions. This research was funded by NIH grants U01 HG009379, U01 MH119509, R01 MH101244, R37 MH107649, and R01 MH109978. S.S.K. was supported by the National Human Genome Research Institute of the NIH under award number F31HG010818. This research was conducted using the UK Biobank Resource under Application 16549.

## Author contributions

S.S.K. and A.L.P. designed experiments. S.S.K. performed experiments. S.S.K., K.D., O.W., C.M-L., and S.G. analyzed data. S.S.K. and A.L.P. wrote the manuscript with the assistance from K.D., O.W., C.M-L., and S.G.

## Competing interests

The authors declare no competing interests.
