## [Peer Review File · Nature Communications]

Reviewer #2 (Remarks to the Author):

The authors have done a number of useful additional analyses and made changes in response to my previous comments. I have only two minor further comments:

1. I think I know understand that when they say "uniquely informative" the authors mean "conditionally informative" or "informative even after considering all other annotations". Assuming I'm right, I'd suggest changing to something along those lines. I interpreted "uniquely informative" to mean informative in a way that nothing else is, but in the response to previous review the authors made clear that's not what they mean. This is somewhat pedantic, but I do think the usage will be confusing to other readers.

2. It's interesting to see the increased emphasis on "pervasive variant-level overlap", which I don't actually think is necessarily the case. My original interpretation of the findings was that the most likely explanation is that certain properties that are enriched in Mendelian variants are also enriched in common disease causing variants. And that thus the scores explain more heritability without necessarily being based on the same variants that cause Mendelian disease. I think this has to be right because most Mendelian causing variants are too rare to be captured in the heritability analysis. This could again be a matter of terminology, where the authors mean simply what I'm saying, that variants that cause Mendelian disease and those that affect common disease share properties.

Reviewer #3 (Remarks to the Author):

The authors have done an excellent job in addressing my previous comments. This is a very well written and comprehensive study.

Reviewer #4 (Remarks to the Author):

The proposed work has provided a novel and informative way to utilize and improve on currently available function annotation scores. The topic of the manuscript is well-suited for the audience of the journal.

The updated manuscript has made substantial improvements with additional analyses and discussions. The authors have addressed all my comments and concerns in this revision. Additionally, I really appreciate the amount of details provided by the authors to explain their revision.

Response to reviewers for NCOMMS-20-10102A (Kim et al.)

Reviewer #2:

The authors have done a number of useful additional analyses and made changes in response to my previous comments. I have only two minor further comments:

We thank the reviewer for acknowledging the additional analyses and other changes in response to previous comments. The two minor further comments are addressed below.

1. I think I know understand that when they say "uniquely informative" the authors mean "conditionally informative" or "informative even after considering all other annotations". Assuming I'm right, I'd suggest changing to something along those lines. I interpreted "uniquely informative" to mean informative in a way that nothing else is, but in the response to previous review the authors made clear that's not what they mean. This is somewhat pedantic, but I do think the usage will be confusing to other readers.

We agree that “unique informative” might be confusing to readers. We replaced “uniquely informative” to “conditionally informative” in all instances. We also made this point clearer in the Introduction section. Changes are highlighted in red font.

2. It's interesting to see the increased emphasis on "pervasive variant-level overlap", which I don't actually think is necessarily the case. My original interpretation of the findings was that the most likely explanation is that certain properties that are enriched in Mendelian variants are also enriched in common disease causing variants. And that thus the scores explain more heritability without necessarily being based on the same variants that cause Mendelian disease. I think this has to be right because most Mendelian causing variants are too rare to be captured in the heritability analysis. This could again be a matter of terminology, where the authors mean simply what I'm saying, that variants that cause Mendelian disease and those that affect common disease share properties.

We agree with the reviewer interpretation that Mendelian variants and common disease variants share common properties. We have thus replaced “pervasive variant-level overlap” to “implying that Mendelian disease variants and common disease variants share similar properties” for clarification in Abstract and Discussions section.

Reviewer #3:

The authors have done an excellent job in addressing my previous comments. This is a very well written and comprehensive study.

We thank the reviewer for indicating that we have done an excellent job in addressing the previous comments, and that the paper is very well written.

Reviewer #4:

The proposed work has provided a novel and informative way to utilize and improve on currently available function annotation scores. The topic of the manuscript is well-suited for the audience of the journal. The updated manuscript has made substantial improvements with additional analyses and discussions. The authors have addressed all my comments and concerns in this revision. Additionally, I really appreciate the amount of details provided by the authors to explain their revision.

We thank the reviewer for indicating that our revised manuscript is substantially improved.